# Examining HIV Knowledge and Sexually Risky Behaviors among Female Sex Workers in Kampala, Uganda

**DOI:** 10.3390/ijerph21020163

**Published:** 2024-01-31

**Authors:** Jude Ssenyonjo, Colleen Mistler, Tanya Adler, Roman Shrestha, Peter Kyambadde, Michael Copenhaver

**Affiliations:** 1Department of Allied Health Sciences, University of Connecticut, Storrs, CT 06269, USA; tanya.adler@uconn.edu (T.A.); roman.shrestha@uconn.edu (R.S.); michael.copenhaver@uconn.edu (M.C.); 2Division of Prevention and Community Research, Department of Psychiatry, School of Medicine, Yale University, New Haven, CT 06520, USA; colleen.mistler@yale.edu; 3Most At-Risk Populations Initiative—MARPI, Department of Allied Health Sciences, University of Connecticut, Storrs, CT 06269, USA; kyambex@gmail.com

**Keywords:** HIV, AIDS, drug use, sexually transmitted diseases, alcohol, female sex workers, FSWs

## Abstract

HIV incidence remains alarmingly high among female sex workers (FSWs) in Uganda, necessitating targeted interventions. This study aimed to identify individual and provider-level barriers and facilitators to primary HIV prevention among FSWs in an urban setting. Focus groups involving FSWs and healthcare providers (HCPs) were conducted to inform the development of tailored prevention interventions. Results revealed that all participants had mobile phones, recent sexual activity, and a history of HIV testing, with high rates of sexually transmitted infections and varying HIV test results. FSWs displayed a well-informed awareness of HIV transmission risks, emphasizing the threat for those not testing positive. They unanimously recognized the crucial role of HIV testing in informing, mitigating risks, promoting ART use, and endorsing consistent condom usage. Despite heightened awareness, HCPs noted potential underestimation of vulnerability. Various challenges, including inadequate condom usage, substance abuse, and client dynamics, underscored the complexity of safeguarding against HIV transmission among FSWs. Widespread alcohol and drug use, including marijuana, kuber, and khat, served as coping mechanisms and social facilitators. Some FSWs successfully reduced alcohol intake, highlighting challenges in addressing substance use. FSWs preferred group discussions in health education programs, emphasizing peer interactions and the effectiveness of visual aids in HIV prevention education. This study provides comprehensive insights to guide the development of targeted interventions addressing the multi-faceted challenges FSWs face in HIV prevention.

## 1. Introduction

The exchange of sexual services, performances, or activities for cash, products, or other forms of payment is referred to as “sex work.” Prostitutes, escorts, strippers, and other professionals in the adult entertainment sector are examples of people who work in the sex industry. Female sex workers (FSWs) around the world continue to bear disproportionate adverse effects associated with HIV [1], with a profound impact on their health and well-being [2,3,4]. The prevalence of HIV among FSWs is staggering, with estimates revealing an overall prevalence of 12% in low and middle-income countries and 30% in countries with high background prevalence [5]. Uganda stands out with a harrowing prediction of 33% of FSWs being infected with HIV, surpassing the national prevalence rates [6]. This bleak reality persists despite the existence of HIV prevention interventions that are struggling to gain traction.

The persistence of HIV transmission among FSWs underscores the presence of complex structural impediments that hinder healthcare access and exacerbate the social and environmental factors that increase their vulnerability [7,8]. Research is progressively revealing the complex interplay of factors at various levels—individual, societal, structural, and policy—that impede HIV prevention initiatives and hinder the essential uptake of testing among FSWs [9,10]. These factors include the alarming infrequency of condom use [11,12], which is driven by economic, social, and interpersonal relational issues such as poverty, partner refusal, trusting that their regular clients do not have STIs [13], and substance use [14,15]. Nonetheless, research consistently underscores the significance of additional individual-level factors, such as advanced age, limited educational attainment, and a lack of comprehensive HIV-related knowledge [16]. Studies have also documented voluntary condomless sex, especially when clients offer FSWs higher incentives [17].

The situation in Uganda is further compounded by the criminalization of sex work and its prevailing stigma [18]. These factors not only adversely affect sex workers’ health and safety but also infringe upon their fundamental human rights, leading to heightened risks of violence, poor sexual health, and HIV/STI infections [19,20,21,22,23]. The criminalization of sex work is a formidable barrier to negotiating condom use and accessing essential health services, including HIV/STI testing and treatment [24,25].

Despite these formidable obstacles, a growing body of evidence highlights the effectiveness of behavioral intervention programs for FSWs [26,27,28]. Routine STI and HIV screening services are pivotal in primary prevention, care, and HIV transmission reduction through timely antiretroviral therapy [29]. However, behavioral, and biomedical interventions, while beneficial, have only yielded moderate community-level results, prompting a call for a combined HIV prevention strategy that includes structural interventions [20,22]. Efforts are needed to provide ART and condoms to FSWs in areas where criminality and stigma impede access to essential services.

The predominant strategy for increasing STI and HIV screening among FSWs in Uganda involves community-based testing through night-time outreach, but it falls short due to limited services and issues of privacy and confidentiality [24,25].

Given the alarming rates of HIV exposure among FSWs, it is imperative to gain insights into their attitudes toward HIV risk, behaviors, and outcomes to design and implement successful preventive programs. To achieve this, we conducted focus groups with FSWs and HCPs in various locations across Kampala, Uganda. These discussions shed light on the complex landscape of HIV risk and prevention, providing essential information to inform future interventions and support this vulnerable population in their quest for improved health and well-being.

## 2. Methods

Between February and March 2023, 6 focus group discussions (FGDs) were conducted. In total, 3 FGDs were conducted with 6–7 FSWs at each session, and 3 separate FGDs were conducted for 6–7 HCPs per session. FGDs were conducted within the grounded theory analytical techniques framework. This study included a total of 20 female sex workers (FSWs) and 20 healthcare providers (HCPs) from five healthcare centers in Kampala, Uganda, as per the approach described by [30,31]. Participant recruitment was facilitated by the Most At-Risk Population Initiative (MARPI), a local organization dedicated to providing services to female sex workers in Uganda. The MARPI recruited the participants with the help of peer educators linked to the communities within the catchment areas around the five health facilities. All participants either lived or worked within the Kampala Metropolitan region. Participants’ eligibility was determined based on the following criteria:Eligibility criteria for female sex workers (FSWs): (a) Minimum age of 18 years; (b) engagement in sex work; (c) ownership of a mobile phone; (d) ability to read and comprehend questionnaires and informed consent forms in either English or Luganda; and (e) availability throughout the study’s duration. Eligibility criteria for healthcare providers: (a) Aged 18 years or older; (b) currently providing services to sex workers under the MARPI; (c) possession of a cell phone; (d) capacity to read and understand questionnaires and informed consent forms; and (e) availability for the entire study period.Identification and recruitment of healthcare providers were conducted in collaboration with the MARPI, utilizing phone calls and word-of-mouth referrals, all with the consent of the respective unit heads. This study specifically targeted clinical officers, nurses, and counselors or social workers who had been directly engaged in the primary prevention of HIV among female sex workers at the MARPI clinic in Mulago and other healthcare facilities over the preceding six months. Healthcare providers were incorporated into the study to offer perspectives on existing facilitators and barriers to HIV prevention among female sex workers.

### 2.1. Data Collection

Following recruitment, participants received written materials explaining the study’s objectives, aims, and methodology. To ensure the smooth execution of the focus groups, participants received reminder calls the day before the scheduled sessions, informing them of the time and location. Two trained MARPI employees facilitated each focus group discussion: one was responsible for conducting and recording the focus groups and the other was responsible for taking detailed notes. Focus groups were conducted in Luganda, a local dialect widely spoken in Uganda. Research staff provided participants with refreshments during the focus group discussions, and a transportation reimbursement of USD 25 was granted to each participant following the conclusion of the evaluation. Confidentiality measures were consistently upheld to safeguard the privacy and anonymity of all participants.

### 2.2. Ethical Considerations

This study adhered to a stringent ethical framework. Ethical approvals were secured from the following institutional review boards: the University of Connecticut Institutional Review Board (IRB)—Approval ID: H21-0024; the AIDS Support Organization (TASO) Research Ethics Committee IRB—Approval ID: TASO-2022-167; and the Uganda National Council for Science and Technology (UNCST)—Approval ID: HS2635ES. These approvals were granted in January 2023. Before their involvement in this study, each participant was required to thoroughly read and sign an informed consent form, which comprehensively detailed the objectives, procedures, and rights of the participants.

### 2.3. Qualitative Data Analysis

The analysis of the qualitative data encompassed two key stages. Initially, participants were invited to respond to open-ended questions to unearth emerging concepts including beliefs, values, attitudes, knowledge, and practices. The research team conducted peer debriefing sessions daily following each focus group session to enhance data rigor and reliability.

The recorded focus groups were transcribed into English, and the resultant transcripts were subsequently imported into the computer-aided qualitative data analysis software MAXQDA Analytics Pro 2022. Within MAXQDA, an analysis team meticulously constructed a codebook, annotated relevant language, and coded the transcripts from the focus group discussions. The thematic analysis of qualitative responses was executed using MAXQDA Analytics Pro 2022, which harnessed qualitative data analysis techniques, including inductive and cross-case analysis [32,33,34,35].

Inductive qualitative data analysis is an exploratory, bottom-up approach that involves extracting patterns, themes, and concepts directly from the data without prior theoretical assumptions. This process aims to generate new theories and insights from the data. The inductive analysis method involves a series of pivotal steps to unveil profound insights from qualitative data, and it was used for this research study. This process began with data familiarization, where research staff fully immersed themselves in the dataset and rigorously scrutinized transcripts and field notes to fully understand the material. Subsequently, open coding was utilized. This procedure disassembled the data into smaller, coherent units, aptly termed codes. These thoughtfully assigned descriptors encapsulated specific phrases, sentences, or paragraphs, facilitating the encapsulation of fundamental concepts, ideas, or recurring themes within the dataset. Once the data were coded, the analysis transitioned into generating categories, wherein research staff amalgamated related codes to create coherent categories or themes. These categories served as a reflection of recurring concepts and ideas that organically emanated from the data. Then, constant comparison ensued, wherein freshly acquired data was systematically contrasted with previously coded information. This continual comparison endeavor was undertaken to refine and expand existing categories, ensuring robust and consistent identification of patterns within the dataset.

Intercoder reliability was rigorously assessed and demonstrated high consistency at 91.43% [36]. This level of reliability attests to the robustness of the qualitative data analysis process and the fidelity of the results.

### 2.4. Quantitative Data Analysis

The statistical software SPSS version 28.0 (IBM Corporation, Chicago, IL, USA, 2017) was employed to analyze socio-demographic data. This facilitated the quantitative examination of key socio-demographic variables, offering complementary insights to the qualitative findings.

## 3. Results

### 3.1. Socio-Demographic Characteristics of Study Participants

This study encompassed a diverse group of participants, including 20 FSWs and 20 healthcare providers, all of whom were of African descent. Among the FSWs, the age range spanned from 23 to 54 years, with an average age of 32 years (standard deviation (SD) 7.159). Healthcare providers exhibited an age range of 22 to 56 years, with an average age of 36 (SD 8.119).

Regarding educational backgrounds, most FSWs reported having received some education, primarily at the primary or secondary levels. However, two FSWs had pursued higher education at college or university levels, while an additional two had not undergone any formal education.

At the time of this study, all FSWs reported possessing a mobile phone, having engaged in sexual activity within the preceding three months, and having undergone an HIV test at least once in their lifetime. Notably, within the three-month timeframe, 95% of FSWs reported receiving a diagnosis of a sexually transmitted infection (STI). Furthermore, 65% had tested positive for HIV, while the remaining 35% had tested negative for the virus. On average, FSWs reported engaging in condomless sexual encounters approximately five times during the study period.

### 3.2. HIV Transmission Awareness

Many female sex workers (FSWs) demonstrated a well-informed awareness of HIV transmission routes. They exhibited a keen understanding that sexual intercourse, mainly when devoid of condom usage, represented the primary avenue for HIV transmission. As one FSW articulated, “I know this virus predominantly spreads through sexual contact. Engaging in sexual activity without protection carries a significant risk of contracting HIV.” Another FSW echoed this sentiment, underscoring the elevated risk of acquiring the virus through unprotected sexual encounters, especially among sex workers who had not yet tested positive for HIV: “For us, the sex workers who have not yet tested positive for HIV, the risk of acquiring the virus is elevated when engaging in unprotected sexual encounters.”

Furthermore, these FSWs placed significant emphasis on the pivotal role of antiretroviral therapy (ART) in HIV treatment, emphasizing the dire consequences associated with non-adherence. 

They also recognized that not everyone who took HIV medication experienced a full recovery, as some individuals faced challenges with drug rejection or improper medication intake, ultimately leading to unfavorable outcomes, including mortality. This awareness further highlighted the detrimental impact of untreated HIV on the immune system, emphasizing the life-threatening nature of the disease. In essence, these insights underscored the gravity of HIV as a potentially fatal condition, particularly when coupled with the risk of co-infection and non-adherence to treatment:

“But then not that everyone that takes that medicine gets well; some have bodies that reject the drugs or even take the medications incorrectly, and in the end, they also cause their death. Then, your immune system becomes very weak. That means that if you don’t take HIV drugs and get infected by other diseases, you may be very affected. So that brings about the fact that HIV is a deadly disease that kills.”

### 3.3. Importance of HIV Testing

Female sex workers (FSWs) unanimously acknowledged the pivotal role of HIV testing in the context of informing individuals about their HIV status, mitigating risky behaviors, initiating antiretroviral therapy (ART) use, and endorsing the consistent use of condoms. They emphasized the critical significance of being aware of their HIV status as a primary means to curtail the transmission of the virus. One FSW articulated this importance by stating the following: 

“It’s essential for someone to undergo HIV testing to ascertain their HIV status. When you’re unaware of your HIV status, the risk of inadvertently infecting numerous people exists. However, upon testing and discovering that you have HIV, you can initiate medication to combat the virus and bolster your immune system.”

Another FSW participant emphasized how HIV testing not only facilitated personal knowledge but also encouraged protective measures, saying, “It helps us to be aware of our own HIV status, adhere to medication, and consistently use condoms, thereby minimizing the risk of transmitting the virus to others.”

The sentiment of potential peril for those who remain unaware of their status was recurrent, with one individual highlighting, “If someone remains ignorant of their HIV status, they are constantly in jeopardy, lacking the awareness necessary to make informed choices regarding their health and interactions.”

Lastly, one FSW shared a personal testament to the transformative power of HIV testing, remarking, “Yes, it indeed prompts you to change your behavior. Personally, the last time I got tested, it motivated me to adopt protective measures at any cost.” These collective insights underscored the profound influence of HIV testing as a catalyst for knowledge, behavioral change, and prevention within the FSW community.

### 3.4. HIV Risk Perception

Female sex workers (FSWs) exhibited a heightened awareness of the risks associated with unprotected sexual encounters and the pivotal role of condom usage in mitigating the transmission of HIV. As one FSW succinctly said, “My understanding is that this virus primarily spreads through sexual activity. When we partake in unprotected intercourse, the risk of contracting HIV is significantly elevated. For us, sex workers who have not yet been diagnosed with HIV, there’s a substantial risk of acquiring the virus if we engage in unprotected sexual encounters.”

However, it is worth noting that some healthcare providers believe that FSWs may sometimes underestimate their vulnerability to HIV due to their risky behaviors. One healthcare provider shared their perspective, saying the following:

“Sometimes, I wonder about their thought process. I believe that their behaviors expose them to the risk of neglecting the potential consequences of contracting HIV. It seems they prioritize their immediate desires over taking preventive measures or gaining a deeper understanding of the ramifications of HIV infection. Their primary focus appears to be on fulfilling their immediate needs.” 

This observation sheds light on the complex dynamics within the FSW community, where risky behaviors sometimes take precedence over preventive measures.

### 3.5. HIV Risk Factors

In the realm of HIV risk factors among FSWs, several vital elements were prominent. Inadequate condom usage, substance abuse, peer influence, financial necessity, and client demands played significant roles in shaping these risks. For some FSWs, the lure of higher pay led them to engage in unprotected sexual encounters, while others found themselves coerced or subjected to condom sabotage.

One FSW candidly described the economic incentives that sometimes swayed their decision, stating, “There are instances where a client may offer more money if you forego condom use, and in situations where you have substantial financial needs, you might find yourself accepting the offer.”

Another participant highlighted the challenges they faced when clients resisted condom use, revealing, “Many men feign intoxication, claiming ignorance about the whereabouts of condoms. When you present your own, they might insist on not using them and pretend they have their stash, even if it’s not the case. This can lead to confrontations and a refusal to use condoms.”

Beyond these issues, additional risk factors included needle sharing, the allure of temptations, and the accessibility of PrEP and PEP (pre-exposure prophylaxis and post-exposure prophylaxis, respectively). These multifaceted risk factors underscore the complex interplay of economic pressures, client dynamics, and the challenges inherent in safeguarding against HIV transmission among FSWs.

### 3.6. Alcohol and Drug Use

Findings from this study revealed that alcohol and drug consumption were widespread practices among FSWs, with many resorting to these substances to bolster their sexual confidence or to navigate the challenging circumstances they encountered. As one FSW explained, “There are times when we turn to alcohol due to our circumstances. You might encounter a client you don’t like or whose situation is unfavorable, and you may need alcohol before engaging with them.”

Various motivations for substance use were identified, spanning from the necessity to stay alert during nocturnal encounters, combat the cold, and yield to peers’ influence. Notably, FSWs preferred substances of choice included marijuana, kuber, and khat. This discovery underscores the intricate and multi-faceted role of alcohol and drug use in the lives of female sex workers (FSWs). These substances often serve as coping mechanisms, performance enhancers, and even facilitators of social interactions, all tailored to their unique circumstances.

### 3.7. Intention to Reduce Substance Use

The discussion around FSWs’ intention to reduce substance use revealed significant challenges in addressing drug and alcohol consumption. Nearly half of those using drugs encountered difficulties in moderating their intake. Nevertheless, there were instances where some FSWs successfully reduced their alcohol intake or abstained from it entirely.

## 4. Proposed HIV Intervention

### 4.1. Preferred Learning Style

When discussing their views about the most effective ways to convey HIV prevention, female sex workers (FSWs) displayed a distinct preference for group discussions within the context of health education programs, underscoring the significance of peer interactions as a valuable component of the learning process. Additionally, they expressed the utility of visual aids and graphics as effective teaching tools to enhance the educational experience.

One FSW articulated the benefits of group sessions, stating the following: 

“I believe these group meetings are quite beneficial. Being part of a group is particularly advantageous because you might come here without knowledge about certain aspects of your life. However, participating in discussions with your colleagues leaves you with newfound insights and knowledge you might not have otherwise acquired.” 

This highlights the collective learning and supportive environment potentially fostered by group discussions, making them a preferred approach for health education among FSWs.

### 4.2. Technology Usage 

In our investigation of the technology utilized by female sex workers, we discovered that most of them owned cell phones, with smartphones being the most common choice. However, they faced obstacles such as limited phone data and literacy issues. Healthcare providers predominantly used phone calls as their main form of communication with FSWs.

One participant noted the advantage of direct phone calls, explaining, “Direct phone calls are the most effective means of communication because even those who own smartphones might not have the financial means to purchase data for viewing text messages or other forms of digital communication.” Given the potential barriers associated with data limitations and digital literacy, this insight underscores the practicality of phone calls as a communication method.

### 4.3. Best Engagement Time and Session Duration

During discussions about the ideal duration for group sessions, healthcare providers presented differing opinions on the best engagement times, proposing both morning and afternoon sessions. Meanwhile, female sex workers (FSWs) expressed a preference for more extended sessions lasting 2 to 3 h. Providers tended to favor shorter sessions, typically 20 to 45 min. This disparity underscores the importance of striking a balance between the FSWs’ desire for deeper engagement and the practical constraints favored by healthcare providers.

### 4.4. Ensuring FSWs’ Participation

In conversations about motivating FSWs to engage fully in HIV prevention programs, the significance of positive attitudes and supportive behavior from healthcare providers emerged as crucial factors. One healthcare provider highlighted the impact, emphasizing that these elements were pivotal in encouraging participation among female sex workers: “These individuals are quite sensitive, as they already grapple with challenges. Encountering a healthcare provider with a compassionate attitude can positively impact their willingness to return for further sessions. Therefore, it is crucial for us to maintain a positive and empathetic demeanor when working with or attending to these individuals.”

Additional strategies to encourage participation encompassed the provision of essential supplies, community outreach efforts, assistance with transportation, ensuring privacy, and actively involving peer educators. Collectively, these comprehensive approaches create a conducive and supportive environment for FSWs to engage in HIV prevention programs.

### 4.5. Attendance Obstacles

In an effort to comprehend the obstacles impeding attendance at HIV prevention sessions, a multitude of barriers were identified by the FSWs. These included transportation challenges, pervasive stigma, financial limitations, and unexpected family emergencies. Further hurdles mentioned encompassed hangovers, literacy constraints, and time restrictions. One participant elaborated on these challenges: “Some of us are apprehensive about attending these sessions because we fear encountering individuals we may know. Therefore, establishing dedicated spaces exclusively for sex workers, free from the presence of others, could greatly bolster our confidence and participation.” This insight highlights the potential benefits of creating safe and discreet environments tailored to the unique needs of sex workers.

### 4.6. Perceptions about Peer Educators

In this study, facilitators examined the degree of trust established by peer educators among female sex workers. The majority of FSWs expressed a strong preference for peer educators, highlighting their trustworthiness and consistent support. As one FSW aptly put it, “The truth is that our peer is exceptional; she embodies all the qualities of a reliable peer. She is a true friend to everyone. Whether in difficult situations or even in legal trouble, we can always count on her support.” 

Furthermore, another FSW emphasized the accessibility and assistance that peer educators facilitated in accessing essential health services, noting, “Whenever I encounter any problem or health challenge, they can swiftly connect me with healthcare providers to ensure I receive the necessary services.”

Nevertheless, some FSWs disagreed about sharing their personal information with peer educators. One FSW expressed her willingness to disclose some information while maintaining certain boundaries, explaining, “I would be open to sharing with them, but to a limited extent; there are some things I might not reveal.” These differing viewpoints underscore the significance of building trust and rapport between peer educators and FSWs to ensure the effectiveness of peer-driven interventions.

## 5. Perceptions from Healthcare Providers

### 5.1. PrEP Knowledge

During discussions about pre-exposure prophylaxis (PrEP) knowledge, healthcare providers showcased a strong grasp of PrEP and its application among high-risk groups, especially within FSWs. They recognized the crucial role that PrEP plays in HIV prevention. One healthcare provider concisely conveyed, “My understanding of PrEP is that it refers to medication prescribed to individuals engaged in high-risk behaviors to safeguard them from HIV infection. In practice, I provide this medication to HIV-negative clients at substantial risk of contracting the virus.” This insight underscores the healthcare providers’ competence in recognizing the significance of PrEP and its targeted use in HIV prevention for at-risk populations.

### 5.2. Intervention Delivery Approach

To determine the most effective method for delivering HIV information during FSWs’ group sessions, healthcare providers leaned towards incorporating both group and individual sessions in their intervention strategies. They emphasized the importance of catering to various personalities within these sessions, highlighting that group interactions could often be more impactful. One provider expanded on this: “People have diverse personalities, and just like in a classroom setting, some may be hesitant to participate, while others are more vocal. In group sessions, it’s often beneficial to allow the participants to guide the discussion.”

Additionally, healthcare providers advocated for open interaction as a valuable approach, where participants actively engage and share their perspectives. One healthcare provider explained, “I find open interaction to be a valuable method. For instance, when discussing topics like PrEP, you can start by asking them about their understanding of PrEP. This allows them to express their misconceptions, and you can subsequently provide them with accurate information.”

Moreover, they proposed innovative strategies such as peer-to-peer models and mobile health initiatives to enhance outreach efforts. They also expressed a willingness to be flexible in accommodating the schedules of female sex workers when conducting sessions at healthcare facilities. As one provider emphasized, “From my experience, the peer-to-peer model has proven to be effective, as reaching FSWs or MSM (men who have sex with men) in their locations is challenging without a trusted intermediary. Therefore, the peer-to-peer model is a successful approach.” These insights underscore the adaptability and creativity of healthcare providers in tailoring their interventions to meet the unique needs of their target populations.

### 5.3. PrEP Services

Regarding awareness of PrEP services, healthcare providers expressed apprehensions regarding its safety, daily adherence, and potential side effects. Among the primary concerns raised was the impact of PrEP on the liver and the possibility of long-term health complications. One provider specifically highlighted, “They asked me if the drugs would affect their liver and if they might face health issues later in life.”

In response to these concerns, providers recommended several strategies. First, they emphasized the need to enhance awareness about PrEP to address misconceptions and knowledge gaps among high-risk populations. Additionally, they advocated for the availability of long-acting injectable (LAI) options, which many individuals found more appealing as they eliminated the daily pill burden. One provider explained, “The key concern I’ve encountered is that individuals are eager for injectable PrEP, as they feel less burdened than daily pill regimens. We should prioritize the introduction of injectable PrEP as it aligns with their preferences and expectations.” Furthermore, they recommended retraining healthcare workers to address these concerns and offer accurate information, underlining the importance of continuous education and support in promoting PrEP use.

### 5.4. Implementation Challenges

During discussions about HIV implementation hurdles, healthcare providers faced numerous challenges, including issues with intoxicated clients, misunderstandings among peers, logistical obstacles, and a general lack of awareness about PrEP. In response, they proposed recommendations to tackle these issues and improve the implementation of their interventions. One pressing challenge revolved around the alcohol consumption of their clients, which hindered effective communication. Healthcare providers explained how clients often arrive at their designated meeting places under the influence of alcohol, rendering educational sessions less effective. One provider explained the following: 

“Alcohol consumption has presented a significant challenge for healthcare workers. We often arrive at hotspot locations to meet with our clients and find them inebriated. The peer may have called them, or the brothel manager may have summoned them for their services. They sit down, and when we attempt to convey information about safer sex practices, some are nodding off, some are barely comprehending, and some are even disruptive. The influence of alcohol from the previous night or the present moment impedes their ability to grasp the information we’re providing. We might spend two hours speaking to individuals who struggle to comprehend our words. Consequently, alcohol has presented an ongoing challenge for us healthcare workers.”

In light of these challenges, healthcare providers recommended the implementation of strategies aimed at reducing alcohol use, raising awareness about PrEP, and overcoming logistical obstacles. They recognized the need for interventions that addressed the alcohol-related impediments to communication and the gaps in knowledge about PrEP within their target populations. By doing so, they hoped to enhance the effectiveness of their outreach efforts and ensure that the information provided would be more readily understood and absorbed.

## 6. Discussion

The findings from our study shed light on the complex web of factors influencing HIV risk and outcomes among FSWs in Kampala, Uganda. The outcomes of this study highlight a significant portion of FSWs with a commendable grasp of HIV-related knowledge, particularly regarding the availability of antiretroviral medication (ART) and HIV prevention strategies. Additionally, 86% of respondents exhibited a heightened awareness of their susceptibility to HIV. Paradoxically, despite this enhanced awareness, the consistent underutilization of condoms persists, influenced by a complex interplay of factors. These include the perpetuation of myths and misconceptions, the challenges associated with alcohol and drug addiction, as corroborated by previous research ^14, 15^, and the compelling impact of financial constraints that frequently lead them to reluctantly forgo the protection of condoms in exchange for higher earnings even when they express intention to do so [37]. On the other hand, Patel et al. present a contrasting perspective by revealing a connection between financial stability and adopting safer sexual practices among FSWs [38]. 

Together, these findings underscore the pressing requirement for economic empowerment programs and skills training tailored to women involved in sex work. In the broader scope, these initiatives have the potential to establish alternative sources of income, thereby alleviating the economic pressures that presently compel them to forgo condom use in their pursuit of higher earnings. The misconceptions uncovered in this study frequently correlate with lower levels of education, a connection well-supported by prior research that consistently demonstrates a strong link between educational attainment and condom usage [39,40,41].

In addition, limited condom use emerged as a significant risk factor among FSWs, with economic incentives and peer pressure contributing to this behavior. A notable proportion reported frequent alcohol and drug consumption, with the majority primarily using marijuana, khat, and kuber. Consistent with prior research, parallel studies have established a correlation between alcohol consumption and heightened involvement in risky sexual behaviors, including having multiple partners and an increased likelihood of engaging in unprotected sexual intercourse [41,42,43,44]. Furthermore, it is linked to an increased prevalence of HIV and sexually transmitted infections (STIs) [45,46], as well as a decreased rate of HIV treatment uptake and adherence [47,48]. Strategies aimed at reducing HIV risk among FSWs should address these behavioral factors. This includes interventions emphasizing the importance of consistent condom use and programs targeting substance use disorders.

The prevalence of drug use among FSWs is a critical concern. Consistent with other studies, a notable proportion reported frequent alcohol and drug consumption, with the majority primarily using cannabis (marijuana), khat, and kuber [49]. While the health effects of khat (miraa) are comparatively less explored than those of cannabis and alcohol, a few studies have indicated potential dependence associated with heavy khat use [50] and an elevated risk of psychosis [49,51]. Considering these associations, it is imperative that future interventions consider the prevalence of poly-substance use among FSWs, as many have reported the harmful consumption of multiple substances. This underscores the pressing need for substance abuse interventions tailored to the specific context of FSWs, including their working hours and the economic factors that drive drug use.

The finding that healthcare providers regularly engage in discussions about drug use with FSWs and offer advice is promising. It is crucial for interventions to continue promoting these discussions and providing support to FSWs who wish to reduce or quit substance use. This proactive approach is instrumental in addressing this population’s complex issues surrounding substance abuse. The preferred learning style among FSWs, such as group discussions and visuals, aligns with existing educational strategies. Healthcare providers’ agreement with group counseling as the preferred method suggests an opportunity to leverage existing preferences for education. Additionally, the acknowledgment of the importance of peer educators is consistent with prior research, as they are often seen as credible and trusted sources of information and support within the FSW community.

The role of technology in communication should not be underestimated. Mobile health (mHealth), which encompasses mobile applications, SMS services, and other digital tools and peer education strategies, has shown promise in increasing the uptake of STI services in facilities [21,24], although research on their effectiveness in preventing HIV among high-risk groups, specifically FSWs, remains scarce. The high rate of cell phone ownership among FSWs indicates the potential for technology-based interventions. However, it is essential to address issues related to phone data troubles and low literacy levels, which may hinder the effectiveness of technology-based interventions. The preferences for afternoon sessions and sessions lasting between 20 and 45 min are practical insights for designing intervention programs. Considering night-time education with peer assistance may address FSWs’ working hours and availability. Flexibility in session timing and duration can help maximize participation.

Motivating factors identified by FSWs include positive healthcare provider attitudes, time spent with facilitators, and the availability of supplies. Transportation facilitation is a significant concern, emphasizing the need to address logistical barriers to participation. Involving healthcare providers and conducting community outreach were also highlighted as valuable strategies. This aligns with comprehensive interventions involving multiple stakeholders and considering individual and structural factors.

Transportation emerged as the primary impediment to attendance, underscoring the necessity for pragmatic solutions to enhance the accessibility of HIV prevention services. The range of barriers discussed, including stigma and financial constraints, underscores the imperative of implementing comprehensive intervention strategies to tackle individual and systemic challenges. These results align with parallel studies conducted among FSWs in different regions of Uganda. The presence of stigma and discrimination from healthcare providers, insufficient availability of user-friendly services, transportation limitations hindering access to health clinics, and low literacy levels collectively act as deterrents, impeding not only women in general but also FSWs from effectively engaging with HIV prevention and treatment services [42,52].

The awareness and knowledge of PrEP among healthcare providers are critical for its successful implementation. Their willingness to provide PrEP to high-risk groups is a positive sign. The concerns raised by healthcare providers, such as safety and adherence, should inform strategies for PrEP delivery. Healthcare providers identified various challenges in implementing HIV prevention programs, including peer misunderstanding, alcohol abuse, and criminalization of sex work. Similar findings have been reported by Matovu et al. [53]. These challenges require tailored solutions and partnerships with relevant stakeholders, including law enforcement and substance abuse treatment centers. Additionally, addressing the availability and accessibility of HIV testing kits is crucial to ensure that FSWs have consistent access to these services.

Our study’s results should be considered in the context of limitations and strengths. Regarding limitations, our study is constrained by a small sample size, potentially limiting the generalizability of findings to a broader population of FSWs. Secondly, the inclusion of FSWs living with HIV exceeding 50% of the sample may introduce bias, as these individuals might possess greater familiarity with HIV risk factors and preventive measures compared to their counterparts not living with HIV. Thirdly, the requirement that all FSWs had undergone HIV testing in the three months preceding the study may introduce selection bias, as participants might represent a subgroup more inclined to engage with healthcare providers and actively seek HIV prevention information, potentially not fully reflecting the diversity of FSWs in the study setting. Lastly, it is essential to acknowledge that our qualitative approach, which delved into the perceptions and experiences of deliberately selected FSWs and healthcare providers, may constrain our findings’ generalizability. Nevertheless, we undertook measures to enhance the external validity of our study by engaging with healthcare providers from public and private healthcare facilities across five divisions in Kampala. We firmly believe that the insights gained from this research may, to some extent, reflect the sentiments and viewpoints of a broader cross-section of Ugandan FSWs and the healthcare providers who cater to their needs.

Despite these acknowledged limitations, our study is valuable to the field. Previous research primarily concentrated on exploring the experiences and perceptions of FSWs when seeking healthcare, mainly overlooking the solicitation of insights for designing tailored interventions. While some sentiments expressed in our study may resemble those captured in prior research, our unique approach lies in the corroboration of these findings through focus groups conducted among healthcare providers. Consequently, our study offers a more comprehensive view of the insights shared by FSWs and partially validates the perspectives articulated by the healthcare providers, representing a form of data triangulation that enhances the credibility and depth of our research.

## 7. Conclusions

The findings of this study provide a comprehensive overview of the complexities surrounding HIV risk and outcomes among FSWs in Kampala, Uganda. Effective HIV prevention interventions for this population must be multifaceted, addressing not only behavioral factors but also structural challenges, such as transportation barriers and the criminalization of sex work. The preferences and insights shared by FSWs and healthcare providers offer valuable guidance for designing and implementing tailored HIV prevention programs to significantly reduce HIV transmission among this vulnerable population. However, it is essential to collaborate with local organizations and policymakers to address the broader issues, including the legal and social context in which FSWs operate. Recognizing this study’s qualitative nature and limitations—specifically, its dependence on open-ended inquiries—we suggest that more thorough, comprehensive methodologies be used in future research projects. This would enable an exhaustive investigation of missed opportunities for HIV prevention in Uganda, offering a more in-depth and comprehensive understanding of the topic.

## Data Availability

The data presented in this study are available on request from the corresponding author (for privacy, legal, or ethical reasons).

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
