# Peer review of "Examining HIV Knowledge and Sexually Risky Behaviors among Female Sex Workers in Kampala, Uganda"

_ijerph, 2024, doi:10.3390/ijerph21020163_

Round 1

Reviewer 1 Report

Comments and Suggestions for Authors

Dear Editor in Chief, thank you for the opportunity to revise the manuscript "Examining HIV Knowledge and Sexual Risky Behaviors Among Female Sex Workers in Kampala, Uganda" by Jude Ssenyonjo et al. 

In this manuscript the authors present data on a survey conducted among FSW in Uganda and interesting perspectives that could be used to tailor HIV prevention and advocate for improved access to prevention. 

This manuscript present answers from FSW to open questions during interviews. The major limits are the small sample size and the lack of standardized questionnaires to identify missed opportunities for HIV prevention and open issues. Though this strategy possibly granted the opportunity to "let people speak" and identify open issues which might not be perceived by HCP and eventually pave the way for further more rigorous studies involving also analyses on a larger sample size. 

I have some comments which I hope would help strengthening this manuscript.

Abstract.

The abstract does not provide any study results. I would suggest including aims, methods and results of this study in the abstract to improve readibility

Introduction.

Please avoid reference to "seroconversion"

According to people first language, please refer to "people living with HIV" instead "HIV infected" or "HIV positive".

Was discussion conducted in groups or individually when dealing with potentially sensitive topics such as FSW personal HIV status?

In which language was it conducted? As "statements" are reported in English.

Methods.

It is unclear to me how many meeting were conducted with each FSW. Please include data on attendance to meetings.

Please include more details on roles of HCP (e.g. doctors, nurses or others) included in the study.

Results. 

Lines 196-200, 208-211, 217-219, 221-222, 236-240 etc, when giving statements, should be italicized. 

Statement 236-240 appears to be stigmatizing said by an HCP caring for FSW. The concept of HCP stigma could be addressed in the study conclusions. 

Please include also the view on PrEP and its uptake by FSW and not only FSW.

Please include viewpoints of FSW on HIV testing. 

Limitations. Please include as study limitation than more than 50% of FSW were living with HIV, and therefore might be more familiar with risk factors and preventive measures, compared to FSW not living with HIV. Moreover, all FSW conducted HIV testing in the 3 months before interview: people participating in this study might not fully represent FSW of the study setting but merely people more eager to engage with HCP and keen to prevent HIV and diffuse knowledge on HIV risk. 

Please acknowledge the small sample size as a study limitation.

The authors might acknowledge that given the qualitative approach of this study and its limitation (open questions) it could pave the way for more rigorous approaches on a larger scale to identify missed opportunities for HIV prevention in Uganda. 

Overall the manuscript is well written, and from a social perspective, interesting. 

Comments on the Quality of English Language

Minor editing of English language required

Author Response

Abstract.

The abstract does not provide any study results. I would suggest including this study's aims, methods, and results in the abstract to improve readability.

  • Thank you for bringing this to our attention; we have updated the abstract accordingly.

Introduction.

Please avoid reference to "seroconversion."

  • We agree with the concerns of the reviewer. We have removed “seroprevalence” from the introduction and left it as “overall prevalence – line 37.”

According to people's first language, please refer to "people living with HIV" instead of "HIV infected" or "HIV positive".

  • This is well noted. However, we have not referred to “People living with HIV” and “HIV-infected or HIV-positive people” in our manuscript. We did adjust the third sentence to avoid the term: “HIV infection.”

Was discussion conducted in groups or individually when dealing with potentially sensitive topics such as FSW's personal HIV status?

  • The discussion was conducted in focus groups. However, HIV status was captured from the participant screening/recruitment form. Participants were not required to disclose their HIV status during the focus group discussions, and having HIV was not part of the eligibility criteria.

In which language was it conducted? As "statements" are reported in English.

  • Focus group discussions were conducted in “Luganda,” a local dialect widely spoken in Uganda, and transcribed into “English” using professional transcribers fluent in both languages. We have added this information into the Methods section, lines 113 and 133.

Methods.

It is unclear to me how many meetings were conducted with each FSW. Please include data on attendance to meetings.

  • Thank you for this observation. We have now included data on the number of FGDs conducted among FSWs and HCPs in lines 81-83: “Three FGDs were conducted with 6-7 FSWs at each session, and 3 separate FGD were conducted for 6-7 HCPs per session.

Please include more details on the roles of HCPs (e.g., doctors, nurses, or others) included in the study.

  • We have included the role of healthcare providers in the methods section, lines 100 - 104.

Results. 

Lines 196-200, 208-211, 217-219, 221-222, 236-240, etc., should be italicized when giving statements. 

  • We agree with the reviewer. We have italicized all the statements or quotes in the entire manuscript.

Statement 236-240 appears to be stigmatizing, said by an HCP caring for FSW. The concept of HCP stigma could be addressed in the study conclusions. 

  • The statement “It helps us to be aware of our own HIV status, adhere to medication, and consistently use condoms, thereby minimizing the risk of transmitting the virus to others.” was said by an FSW living with HIV, not a healthcare provider. She made that statement not to stigmatize anyone but to encourage others to test for HIV. We clarified who said in the quote on line 221.

Please include also the view on PrEP and its uptake by FSW and not only FSW.

  • This was not well captured during the focus groups.

Please include FSW's viewpoints on HIV testing. 

  • The statements in lines 236-240 are viewpoints from FSWs who attended the focus groups.
    • “It helps us to be aware of our own HIV status, adhere to medication, and consistently use condoms, thereby minimizing the risk of transmitting the virus to others.”
    • “If someone remains ignorant of their HIV status, they are constantly in jeopardy, lacking the awareness necessary to make informed choices regarding their health and interactions.”

Limitations. Please include as a study limitation that more than 50% of FSWs were living with HIV and, therefore, might be more familiar with risk factors and preventive measures compared to FSWs not living with HIV. Moreover, all FSWs conducted HIV testing in the 3 months before the interview: people participating in this study might not fully represent FSW of the study setting but merely people more eager to engage with HCP and keen to prevent HIV and diffuse knowledge on HIV risk. 

  • Thank you for this contribution. We have incorporated these limitations in the manuscript.

Please acknowledge the small sample size as a study limitation.

  • We agree and have included a small sample among the study limitations.

The authors might acknowledge that given the qualitative approach of this study and its limitation (open questions), it could pave the way for more rigorous approaches on a larger scale to identify missed opportunities for HIV prevention in Uganda. 

  • Thank you for your contribution. We agree and have included this acknowledgment in the manuscript.

Overall, the manuscript is well-written and, from a social perspective, interesting. 

  • We thank you for the compliment and great feedback.

Reviewer 2 Report

Comments and Suggestions for Authors

Dear Authors,

My comments:

1. Please, check all abbrevations if they are described and appear in proper place.

2. I do not understand criterion > 18 y.o. if we talk about health providers.. to be health provider the age is higher...

3. You wrote that participants got money...is it a good idea?

4. Is is great that you wrote about it, but why in your opinion this article is unique?

Author Response

  1. Please check all abbreviations to see if they are described and appear in the proper place.
  • Thank you for this advice. We have checked all the abbreviations and made sure that they are in the right places throughout the manuscript.
  1. I do not understand the criterion > 18 years. if we talk about health providers. to be a health provider, the age is higher.
  • Thank you for this comment. We selected 18 years as the minimum age for all participants, whether female sex workers or healthcare providers.
  1. You wrote that participants got money...is it a good idea?
  • Thank you for this question. Yes, we found it necessary to compensate our study participants for their transport and time spent in the interview, as it is a normal practice in the research fraternity in Uganda and elsewhere in the world.
  1. It is great that you wrote about it, but why, in your opinion, is this article unique?
  • Thank you for this question. This article is unique in providing direct insights from FSWs and providers regarding HIV prevention barriers and facilitators among our target population. These insights can be used to improve HIV prevention implementation among FSWs.

Reviewer 3 Report

Comments and Suggestions for Authors

The current manuscript sought to understand HIV prevention efforts for female sex workers in Uganda. Focus groups were utilized to elucidate FSWs' and providers perceptions about HIV risk and prevention. The results are interesting, especially when the perceptions between the two groups align. I have a number of questions regarding the methods of the study, however. 

-This is a minor point of clarification, but what do the authors mean by “lower-tier venues” on line 44?

-There needs to be a bit more scaffolding connecting mHealth to community-based testing efforts discussed in lines 61-67. As someone who is only peripherally familiar with the related literature, this would help. 

-More detail needs to be provided regarding recruitment efforts and MARPI.

-How long was the study period?

-I am a bit confused—were data collected via focus groups or individual interviews? 

-It might be helpful to define sex worker as this can include a range of activities. 

-The connection between location of HIV testing and birth of children with HIV between lines 183-188 is confusing. This needs a more detailed explanation. 

-This is another minor point: quotes are presented inconsistently in italics. 

-Given the space devoted to mHealth and community-based interventions in the introduction, it would be helpful if more space were devoted to this in the discussion. 

Author Response

The current manuscript sought to understand HIV prevention efforts for female sex workers in Uganda. Focus groups were utilized to elucidate FSWs' and providers' perceptions about HIV risk and prevention. The results are interesting, especially when the perceptions between the two groups align. I have a number of questions regarding the methods of the study, however. 

-This is a minor point of clarification, but what do the authors mean by “lower-tier venues” on line 44?

  • Thank you for this question. We have deleted this confusing sentence from the manuscript for clarity.

-There needs to be a bit more scaffolding connecting mHealth to community-based testing efforts discussed in lines 61-67. As someone who is only peripherally familiar with the related literature, this would help. 

  • We agree and have explained a little more about mHealth for clarity.

-More detail needs to be provided regarding recruitment efforts and MARPI.

  • Additional information regarding MARPI’s recruitment efforts has been added – Lines 86-87.

-How long was the study period?

  • Thank you for this question. The study took place between February and March 2023. We clarified this in line 80.

-I am a bit confused—were data collected via focus groups or individual interviews? 

  • The data was collected via focus groups, we removed the term “interviews” from the manuscript for clarity.

-It might be helpful to define sex worker as this can include a range of activities. 

  • We agree and have now defined sex work as follows in the introduction section of the manuscript: “The exchange of sexual services, performances, or activities for cash, products, or other forms of payment is referred to as "sex work." Prostitutes, escorts, strippers, and other professionals in the adult entertainment sector are examples of people who work in the sex industry.”

-The connection between the location of HIV testing and the birth of children with HIV between lines 183-188 is confusing. This needs a more detailed explanation. 

  • We agree and have decided to delete that quote since it appears to be confusing.

-This is another minor point: quotes are presented inconsistently in italics. 

  • We agree and have now revised all the quotes for consistency throughout the manuscript.

-Given the space devoted to mHealth and community-based interventions in the introduction, it would be helpful if more space were devoted to this in the discussion. 

  • We agree and have moved that paragraph to the discussion section.

Reviewer 4 Report

Comments and Suggestions for Authors

Thank you for this interesting paper providing some useful insights into perceptions of HIV risk, and HIV knowledge, of a group of female sex workers in Kampala. It also makes important recommendations for what interventions may be required to protect this group from HIV. The topic is not especially new, but it provides a useful and relatively current insight into the Ugandan context. 

The paper is well written and structured, and I have only a couple of relatively minor comments.

Here are my comments on each section. 

INTRODUCTION

This section provides a helpful and detailed background and justifies why the study was undertaken. You provide sufficient detail of the context, and the current situation. This is useful as this particular topic has much historical research. 

METHODS

This section provides a detailed overview of your methodology and how you gathered and analysed the data. The qualitative analytical process seems thorough.

I note that ethical approval for the study was granted, and that you obtained consent from participants.

Two minor queries:

1.     You state in a couple of places that the respondents were required to read (on one occasion ‘thoroughly’) background information about the study. Were you certain that the information was at a suitable level? Was there extra support for follow up queries from respondents?

2.     Were the health care workers included in the same focus groups as the sex workers, or were these separate events? Sorry if I missed this, but it would be useful to know if you haven't mentioned it. 

RESULTS

These are presented logically and thematically. You use interview excerpts to illustrate the point you are making.

It’s interesting to see why the sex workers make certain decisions, and also the (continuing) vulnerability of this group when (for example) using alcohol to cope with their day-to-day stresses. The perspectives of health care workers are helpful. 

I think the section (line 360) needs an introductory sentence, as this reports exclusively from the perspective of health care providers. This shift requires some kind of bridge from the previous section. 

DISCUSSION/CONCLUSION

You draw on your findings and provide a useful summary, and also discuss how these fit with similar studies elsewhere. You highlight especially the complex interplay of multiple factors that still present HIV risk to sex workers, despite increased awareness and access to (relatively) new interventions, such as PrEP.

You highlight the limitations of your study, which is useful, and your recommendations are valid, based on your findings. 

REVIEWER RECOMMENDATIONS 

1.     Confirm that sufficient support was given to sex workers during the consenting process (answering queries, etc.). 

2.     Clarify if the focus groups combined both sex workers and health care workers or separated them. 

3.     Provide a bridging sentence at line 360 [the shift to perspectives of health care workers only]. 

Author Response

Thank you for this interesting paper providing some useful insights into perceptions of HIV risk and HIV knowledge of a group of female sex workers in Kampala. It also makes important recommendations for what interventions may be required to protect this group from HIV. The topic is not especially new, but it provides a useful and relatively current insight into the Ugandan context. 

The paper is well-written and structured, and I have only a couple of relatively minor comments.

Here are my comments on each section. 

INTRODUCTION

This section provides a helpful and detailed background and justifies why the study was undertaken. You provided sufficient detail of the context and the current situation. This is useful as this particular topic has much historical research. 

Thank you for identifying the strengths of this manuscript.

METHODS

This section provides a detailed overview of your methodology and how you gathered and analyzed the data. The qualitative analytical process seems thorough.

I note that ethical approval for the study was granted, and you obtained consent from participants.

Two minor queries:

  1. You state in a couple of places that the respondents were required to read (on one occasion ‘thoroughly’) background information about the study. Were you certain that the information was at a suitable level? Was there extra support for follow-up queries from respondents?
  • We thank you for this question. Yes, the research staff were present and offered extra support to respondents who needed clarity or additional information about the study.
  1. Were the healthcare workers included in the same focus groups as the sex workers, or were these separate events? Sorry if I missed this, but it would be useful to know if you haven't mentioned it. 
  • The focus groups for healthcare workers were conducted separately from female sex worker’s focus groups. We clarified this in lines 80-83

RESULTS

These are presented logically and thematically. You use interview excerpts to illustrate the point you are making.

It’s interesting to see why the sex workers make certain decisions and also the (continuing) vulnerability of this group when (for example) using alcohol to cope with their day-to-day stresses. The perspectives of healthcare workers are helpful. 

I think the section (line 360) needs an introductory sentence, as this reports exclusively from the perspective of healthcare providers. This shift requires some kind of bridge from the previous section

  • Thank you for this suggestion. We have refined that section to make it clear that those insights were from female sex workers, not healthcare providers.

DISCUSSION/CONCLUSION

You draw on your findings, provide a useful summary, and also discuss how these fit with similar studies elsewhere. You highlight especially the complex interplay of multiple factors that still present HIV risk to sex workers despite increased awareness and access to (relatively) new interventions, such as PrEP.

You highlight the limitations of your study, which is useful, and your recommendations are valid based on your findings. 

REVIEWER RECOMMENDATIONS 

  1. Confirm that sufficient support was given to sex workers during the consenting process (answering queries, etc.). 
  • Yes, we confirm that research staff were present and offered extra support to respondents who needed clarity or additional information during the consenting process.
  1. Clarify if the focus groups combined both sex workers and health care workers or separated them. 
  • The focus groups for healthcare workers were conducted separately from female sex worker’s focus groups.
  1. Provide a bridging sentence at line 360 [the shift to perspectives of health care workers only]. 
  • We have revised that section to make it clear that those insights were from female sex workers, not healthcare providers.

Round 2

Reviewer 1 Report

Comments and Suggestions for Authors

Authors addressed my comments. 

Reviewer 2 Report

Comments and Suggestions for Authors

I accept your reply.

Reviewer 3 Report

Comments and Suggestions for Authors

The authors were responsive to reviewer comments and the manuscript's clarity is improved. Interesting addition to the literature.